# Supporting care engagement in primary care; the development of a maturity matrix

**René Wolters**[1]*, **Ibtissam Mokadem**[1,2], **Michel Wensing**[3], **Jozé Braspenning**[1]

**1** Radboud University Medical Center, Radboud Institute for Health Sciences, Scientific Center for Quality of Healthcare, Nijmegen, The Netherlands, **2** Department of Registration, Netherlands Comprehensive Cancer Organisation, Utrecht, The Netherlands, **3** Department of General Practice and Health Services Research, Heidelberg University Hospital, Heidelberg, Germany

☯ These authors contributed equally to this work.
* rene.wolters@radboudumc.nl

## Abstract

### Background

Care engagement or active patient involvement in healthcare contributes to the quality of primary care, but organisational preconditions in routine practice need to be aligned. A Maturity Matrix for Care Engagement to assess and discuss these preconditions in the general practice team was developed and tested on feasibility and acceptability in general practice.

### Methods and findings

A systematic user-centred approach was applied, starting with a scoping literature search to describe the domains on the horizontal axis of the maturity matrix. The domains and growing steps (vertical axis) were refined by patients (n = 16) and general practice staff (n = 11) in three focus group discussions and reviewed by six experts (local facilitators and scientists). Seven domains could be distinguished: *Personalised Care*, *Shared Decision Making*, *Self-Management*, *Patient as Partner*, *Supportive Means*, *Patient Environment*, and *Teamwork among Healthcare Professionals*. The growing steps described three to six activities per domain (n = 32 in total) that contribute to care engagement. Local facilitators implemented the tool in two general practice teams according to a user guide, starting with a two-hour kick-off meeting on care engagement. In the next step, practitioners, nurses and assistants in each practice indicated their score on the domains individually. The scores were discussed in the facilitated practice meeting which was aimed at SMART improvement plans. Feasibility and acceptability were assessed in interviews showing that the tool was well received by the pilot practices, although the practice assistants had difficulties scoring some of the activities as they did not always relate to their daily work. An assessment after three months showed changes in practice organisation towards increased care engagement.

### Conclusions

The maturity matrix on care engagement is a tool to identify the organisational practice maturity for care engagement. Suggested adaptations must be implemented before large-scale testing.

**Data Availability Statement:** Participants have given their informed consent for the use of anonymized fragments of qualitative data. Participants did not consent to provision of the full raw dataset to persons other than the research

team. Since the raw interviews and transcripts contain sensitive information, even anonymized raw data can compromise their confidentiality. Therefore, current Dutch privacy law and institutional regulations prevent the a priori sharing of the full raw dataset. Considering the importance of data-sharing and providing insight into the research, data access will be considered upon request, evaluating each inquiry individually. Requests for data access may be sent to the department of IQ healthcare of the Radboudumc at iqhealthcare@radboudumc.nl.

**Funding:** This study was funded by an unconditional grant from AQUA-Institut, Göttingen, Germany. (https://www.aqua-institut.de/) The funder played no role in the study design, data collection and analysis, decision to publish, or preparation of the manuscript.

**Competing interests:** The authors have declared that no competing interests exist.

**Abbreviations:** MM-CE, Maturity Matrix Care Engagement.

## Introduction

The active engagement of patients in healthcare practice has been strongly advocated in recent years, both as an ethically desirable aim and a means to enhance patient self-management of health and disease [1–6]. Concepts that apply to this idea are patient co-production of healthcare and patient partnerships in healthcare [5, 7–9], both of which imply a two-way interaction between the patient and healthcare provider. The more substantial engagement of patients requires commitment from all members in a healthcare team in the context of interprofessional collaborative care [10–12]. However day-to-day management of general practice need more insight into the organisational preconditions that must be arranged to implement it in routine practice.

Implementation of care engagement is challenging, particularly in the growing, older population with multimorbidity, and many interdependent health goals [13–15]. Protocol-driven translation of evidence based practice principles in routine healthcare practice is stressed [6, 14–16] as is the allocation of time and other scarce resources [1, 2, 13]. Nevertheless, the involvement of patients in daily care results in better health outcomes [2, 7, 17], better quality of care service [18] and safer health care [2, 3].

Engaging patients in primary care means that patients should become partners in rather than solely recipients of care [7, 9, 19, 20]. Healthcare professionals should be familiar with a patient's personal circumstances by adopting an approach that is respectful, open and supportive, without separating illness from experiences, values, wishes, expectations, needs and activities [21, 22]. From this, achievable, personal goals can be set by both the patient and the healthcare professional, improving health behaviour [21]. Many scientific papers related to this subject are attempting to guide the healthcare professional in their quest to increase mutual care engagement [4, 8, 9, 13, 17, 19–29]. High-quality patient engagement develops most when a team has an awareness of its value and a practice mindset or conscious philosophy that encourages this team to focus on its achievements [1].

Implementing high-quality patient engagement requires interprofessional and integrated care in general practice which covers a multitude of topics, goals and strategies related to care, organisation and systems [27, 29–31]. To implement care engagement in a sustainable manner, it is essential to assess the organisational preconditions and then tailor implementation strategies to the context in which implementation is planned [26, 29, 32–35]. Apart from changes in communication and working on a trusting patient-physician relationship, practitioners also need to involve the relevant parties, and include cultural and structural changes [36]. At first, practices need support to initiate and continue the process [9, 13, 27, 29, 37].

A *Maturity Matrix™* on practice organisation already has been developed for general practice teams including a self-assessment tool and the support of a trained facilitator [38]. On the horizontal axis, the different domains of practice organisation are distinguished, such as managing staff, organisational procedures, using patient data, et cetera. On the vertical axis, the different growth steps are defined with a significant description in each of the cells. The tool facilitates the identification of separate steps to be taken while growing towards the optimal situation; the optimal situation to be aimed for during a certain period can be decided by the practice itself. Moreover, for some domains, a practice team can raise the bar higher than other teams. With such a tool, a practice team can work step-by-step towards their optimal situation. Comparable maturity matrices have been developed for issues other than practice organisation, such as patient safety, dental care or organising primary care networks [39–41]. Such a maturity matrix can be refined to specially stimulate the care engagement at the organisational level.

The aim of this paper is to describe the development of a tool to support activities to optimise the implementation of organisational aspects on care engagement in general practice,

e.g., a maturity matrix. The matrix, as well as the user guide, has been developed systematically with local stakeholders (patients, healthcare professionals and facilitating organisations) to increase the improvement of results [42–44]. The procedure of the development is embedded in a number of sequential steps (stakeholder involvement, literature search, focus group interviews, refinement with stakeholders, user guide). A qualitative evaluation shows first impressions of the feasibility and acceptability of the tool in routine general practice.

## Methods

The development of the matrix is performed in a systematic manner based on literature and stakeholder involvement, pilot-testing and an evaluation comparable to the Medical Research Council guidelines for the development of complex interventions to improve and implement health and healthcare [42]. The participation of local parties took place to promote community capacity building and sustainability as research subjects own the research process, develop skills, and use research results [43]. The project plan formulated by a scientific project team (authors) and their stakeholders (patients, primary care professionals, local primary care facilitator) is presented in the methods section, and the results of the actions undertaken are described in the next section.

### Project design

An explorative user-centred study was designed to develop and test a tool for general practices to support the organisation of care engagement [45]. The involvement of the stakeholders led, at the start of the project, to a refinement of the aim of the tool as follows: (a) to map one's performance in care engagement in general practice, and (b) to elaborate the obtained goals and decide on further actions, as well as how these actions can be organised in such a way that the practice staff feels motivated to perform them. Having set these goals, a scoping literature search was performed on the effect of care engagement on the quality of care to identify relevant actions for care engagement that should be organised. This scientific search was checked and completed in focus groups including patients and general practice staff. Finally, the tool was tested according to an implementation protocol and evaluated for its feasibility and acceptance. The different aspects are described in more detail below.

### Stakeholder involvement

Care engagement was explored with representatives of a regional patient organisation, a local general practice, and a local organisation to facilitate primary care (n = 7). All parties agreed on the need to facilitate the organisation of care engagement in general practice, in which a maturity matrix tool played a central role in provoking the discussion in general practice teams on the organisation of the different actions to improve care engagement. The project plan was approved, and the stakeholders were involved in all phases of the project, including recruitment, development of the tool, the implementation protocol and reflection on the results. In addition, two international experts in the field of care engagement (GE, MM) were interviewed to discuss the procedure followed and the prototype. During the project, it was revealed that the patient organisation could no longer provide a representative due to a shortage of manpower (secondary to long-term illness) within their own organisation. Therefore, the user guide was developed over three meetings with the local facilitating organisation (n = 4) and the scientific project team, including a practicing general practitioner. The local facilitators supported the testing of the tool in some voluntarily recruited general practices, and the scientific team organised a reflective discussion on the results of the whole project, involving the local facilitators (n = 4).

## Literature search on care engagement actions

The scoping literature search was performed to decide upon the number of domains on the horizontal axis, and the growing steps (vertical axis). This search was performed at the start of 2017 in MEDLINE, EMBASE and PsycINFO using the following search terms: "Patient Participation", "Patient Engagement", or "Patient Empowerment", combined with "Ambulatory Care", "Primary Health Care" or "Family Practice". We limited our search to papers in the English language with an abstract available, published between January 2001 and December 2016. The references were imported in a database, and the titles and abstracts of the papers were screened on relevant actions for care engagement. Complete papers were included after positive hits were found. Furthermore, the publications selected were screened for possible relevance to our research question ("snowballing") and completed with literature known to the researchers (JB, IM, RW). Finally, known validated instruments in relation to care engagement were added to the database. All of these elements found in these studies were plotted in a provisionally maturity matrix on care engagement by two authors (IM, RW). In case of discrepancies, there was discussion until agreement was reached on the choice of category. If there was no agreement, a third author (JB) was consulted. In this way, a preliminary prototype was formulated.

## Stakeholder involvement in further development of the prototype

The prototype based on scientific literature was presented for more specific interpretation and prioritisation in three focus groups that were organised into (a) patients with chronic care, (b) representatives of patient interest groups and (c) healthcare professionals from a primary care setting. The recruitment was performed using purposive sampling, accounting for diversity in gender, age and chronic conditions. The patients were gathered from the practice population (n = 4200) of an urbanised village (25,000 inhabitants). A total of 12 were invited by telephone by their own general practitioner; one resigned from the study because of his medical condition, one was not able to make the appointment and one did not show at the time of the focus interview. The regional patient organisation invited participants by social media; six organisations showed interest. These patient interest groups are semi-professional, non-profit organisations partly funded by the government, contributions of members and donations by the public. The care professionals were considered to be a representative sample of professionals working in the region as they were purposely selected to achieve a certain spread in gender, age, years in the profession, geographical area and professional background and were invited by the local organisation to facilitate primary care. All of the invited individuals also took part in the focus interview. The focus group interviews started with an open question on the perceived (or provided) chronic care. Then, more in-depth questions were asked regarding *personalised care*, *abilities of a patient as partner in care*, *support from healthcare professionals*, *patient environment and caregivers* and *teamwork among healthcare professionals*. Eight to twelve participants per group were invited to an accessible location. Every focus group interview was transcribed and coded in Atlas.ti^TM version 8 (ATLAS.ti Scientific Software Development GmbH) by two researchers (IM or RW). A framework analysis compared the codes in the interviews to the themes found in the scoping overview [46]. The allocated codes were compared and were discussed between IM and RW in cases of differences, aiming for consensus; in the case of disagreement, JB was involved. The maturity matrix on care engagement was finalised according to the discussion in the focus group (see Results section). This tool was presented to two international experts in the field of care engagement with special emphasis on methodological issues and to representatives of the regional facilitating organisations for the more practical point of view in day-to-day practice. Their comments were integrated into the Maturity Matrix Care Engagement before the testing in a pilot setting.

| Box 1. Protocol of implementation pilot | |
|---|---|
| **T0:** | A two-hour kick-off meeting on the concepts of "Patient Centred Care" |
| | E-mails were sent to all the members of the primary care team including a link and an invitation to fill out the **MM-CE** |
| | Data from the **MM-CE** were collected into a feedback report |
| **T0 + 1 month**: | A two-hour facilitator meeting presenting the feedback based on the **MM-CE** to the team and facilitating SMART formulated plans for changing daily practice |
| **T0 + 4–6 month:** | Evaluation and reflection |

## The user guide

The user guide was designed by the research team (an organisational psychologist (JB), a general practitioner (RW), an epidemiologist (IM) and two implementation consultants). It was agreed to start with a meeting at the practice location, introducing the concept of patient-centred healthcare to the practice team. In the next meeting, the facilitator discussed the scores on the self-assessment of the maturity matrix with the practice team to decide on further actions and how to perform them (see Box 1). For support, the facilitator remained in contact with the practice team during the months after this meeting. The two meetings were accredited and were of no cost (other than their own time invested) for the participating practices.

## Evaluation and reflection

Two to four practices were considered to test the tool. These practices were recruited by invitation in a two weekly newsletter sent by e-mail to all primary care practices in the region of the local facilitating organisation (n = 54). Field notes and audio recordings were made during both meetings (see Box 1) for evaluation (with the informed consent of all participants). After each session, all participants completed a short, anonymous survey with a total of 11 questions (process and content) and two open questions regarding "tops and tips". Participants and facilitators were asked to report their time invested. The gathered data were analysed with descriptive statistics. Three to four months after the second meeting an interview took place with one representative from each of the participating practices. In this interview, facilitators and barriers and the position towards future developments in the practice were discussed, with regards to care-engagement. As a result of this interview, a report was made and verified by the interviewee.

## Ethics approval and informed consent

The research ethics committee of the Radboud University Nijmegen Medical Centre decided that the study did not fall within the remit of the Medical Research Involving Human Subjects Act (file number 2018–4104). Written informed consent was obtained from all participants. They were aware that participation was voluntarily and that they could withdraw at any point during the study. Standard procedures concerning confidentiality, anonymity and the secure storage of data were maintained throughout.

## Results

### The maturity matrix on care engagement

**Identification of horizontal and vertical axis.**   A total of 38 papers were considered relevant for axis identification. The selection was based on 289 references out of 3,086 unique

papers identified from three databases; snowballing produced another 13 papers, and 15 papers on validated instruments measuring aspects relevant to the subject (i.e., patient activation, perceived patient-physician interaction, impact of health education, shared decision making, self-management, patient assessment of care, caregivers' teamwork) were also added (S1 File). A total of 325 items were extracted from these papers as possible actions contributing to care engagement. Discussions and clustering (by IM, JB and RW) led to seven domains with several activities (items): *personalised care* (27 items), s*hared decision making* (52 items), *self-management* (89 items), *patient as partner* (39 items), *supportive means* (55 items), *patient environment* (12 items) and *teamwork among healthcare professionals* (51 items). The project team decided that the growth steps (vertical axis) could best be represented by different bundles of activities; the more activities in a certain care engagement domain, the stronger the growth.

**Interpretation and prioritisation by patients and healthcare professionals.** The group characteristics of the three focus groups (patients, patient organisation and healthcare professionals) are presented in Table 1. The interviews lasted from 95–128 minutes. Patients prioritised autonomy, accessibility of care and practice, and continuity of care. Patient organisations wanted their caregivers to show more interest in the specific conditions of patients in relation to their environment and to deliver customised care instead of following protocols; they agreed with the patients with the need for continuity of care. The healthcare providers wanted to meet the informational needs of patients and asked for an increase in consultation time, and a decrease in the number of patients per physician.

A total of 916 codes were allocated in the interviews, with 821 codes attributed to elements that had already been found in the scoping literature search. Patients and patient organisations showed most interest in the themes *Patient as Partner* and *Patient Environment*. The healthcare professionals had the most codes in *Personalised Care* and *Patient as Partner*. All themes within the framework of the scoping literature search were identified in quotations of the interviews; no extra themes emerged.

The 95 "new" codes found in the focus groups were more detailed in specifications compared to the themes already found in the framework. They included items such as "continuity of care", "more time per patient", "higher caregiver involvement", "patient as an expert by experience", "flexibility in personalised care versus rigidity of care protocols", "usage of patient peer groups" and "patient autonomy". Overall, it could be concluded that the focus group interviews confirmed the items found in the scoping literature search, and some were reformulated.

**Table 1. Characteristics of the focus group interviews.**

| | Patients | Patient organisations | Care professionals |
|---|---|---|---|
| **Number of interviewees** | 9 (4 male) | 6 (2 male) | 11 (4 male) |
| **Mean age** | 52 years (28–85 years) | 51 years (32–72 years) | 47 years (32–62 years) |
| **Composition focus group** | Patients with diabetes, chronic obstructive pulmonary disease, heart failure, chronic pain, Crohn's disease, cancer, and an informal caregiver | Representing patients with diabetes, thyroid diseases, mental disease, informal caregivers, a regional patient organisation and a patient advisory board. | Five primary care physicians, two primary care physicians with an area of special interest, three practice nurses, one care manager |
| **Duration interview** | 100 minutes | 128 minutes | 95 minutes |
| **Codes found within framework** | 325 | 168 | 328 |
| **Codes found by open coding** | 34 | 22 | 39 |

**Check with experts.**    The presentation of the tool (S2 File) to two international experts in the field of care engagement led to suggestions regarding the methodological approach. They underlined our observation that defining consecutive steps (vertical axis) as in the original maturity matrix on practice organisation [38] was not realistic and supported our definition of discriminative activities. A suggestion was made to visualise the scores in a spider web diagram with a total average and average per domain, eventually in relation to a benchmark. Data provided by the tool should be merely for internal use with the possibility of sharing tools and tips with other practices. The user guide of the maturity matrix for practice organisation in primary care will be suitable. Therefore, a local facilitator will guide the discussion on care engagement based on the self-assessment with the maturity matrix in order to support a SMART improvement plan.

**Feasibility and acceptance.**    Five (5/54) practices initially showed interest in the pilot. Three of these withdrew for various reasons (project did not fit the organisational plan and/or untimely schedule). Two general practices in the region agreed to participate in a pilot (see Table 2 for practice characteristics); these two practices represent the predominant practice form and size in the region and are situated in a large regional town respectively in a village. At the suggestion of the facilitating organisations, a decision was made to complete recruitment and conduct the pilot study in these two practices. All members of the general practice teams (practitioner, nurse, assistant) were invited to take part. The two practice teams included 7 physicians (4 male, average working experience 18 years (7–32 years)), 10 practice nurses (all female, average working experience 17 years (5–35 years)) and 8 practice assistants (all female, average working experience 15 years (0.5–30 years)). In the Netherlands, a practice assistant performs receptionist tasks as well as simple delegated and protocolled medical procedures (such as measuring blood pressure, sampling blood, injecting medication, rinsing cerumen and removing stitches). The kick-off session was well received and considered useful, as it motivated both practices to work with the concept of patient-centred care in the future. The practices realised that there was more to a patient than their illness or condition and that a holistic view may be to the benefit of the patient. They also recognised that interacting in this way may also be beneficial in relation to others, such as colleagues. Both practices were eager to get started and expected a more practical approach. They still felt unclear regarding what they could do to change daily practice, however.

Not all team members were present (n = 21/27). After two weeks, 11 of 19 (with two incorrect mail addresses) surveys were completed; 15 of the 19 surveys were completed after a reminder. Data from the *Lime-Survey* (Lime Survey: An Open-Source survey tool /LimeSurvey GmbH, Hamburg, Germany) were collected and transcribed on a spreadsheet; a feedback report with web diagrams for each theme was generated. Feedback reports showed results that were differentiated to the four functional levels (practice assistant, practice nurse, general practitioner and "other").

**Table 2. Characteristics of participating practices.**

|  | Practice A | Practice B |
|---|---|---|
| **Practice setting** | City (160,000 inhabitants) | Urbanised village (17,000 inhabitants) |
| **Practice size (number of patients)** | 4800 | 4700 |
| **General Practitioners** | 4 | 3 |
| **Practice nurses** | 5 | 5 |
| **Practice assistants**[a] | 4 | 4 |
| **Practice manager** | 1 | 1 |

[a] A practice assistant performs receptionist tasks as well as simple delegated and protocolled medical procedures.

The general practitioners and practice nurses were able complete all or most of the questions. The practice assistants indicated that approximately a quarter of the questions did not apply to their daily work. In particular, questions within the themes "*Shared Decision Making*", "*Supportive Means*", "*Patient Environment*" and "*Teamwork among Healthcare Professionals*".

The teams were very interested in the results of the maturity matrix on care engagement, MM-CE. They recognised the profiles as presented in the feedback reports. There was little discussion in both practices regarding the how and what of the differences between the separate professional groups. The domains of the MM-CE were considered to illustrate the practice situation towards care engagement. The tool was thought to provide good insight into a process that was already going well and provided information on the possibilities for renewal and updating. A major barrier was the difficulty with completing the MM-CE for practice assistants. This was partly due to the abstract language used and partly because of difficulties in relating certain items in the MM-CE to their daily routines.

Both practices needed a practical approach to get off a good start and thought that the work-up provided leads to direct patient care and respects different disciplines. The facilitator session offered the opportunity to work on the themes in a teamwide manner, thereby promoting collaboration and deriving a joint solution. Participants were challenged to formulate ideas to increase the practice populations' well-being. These ideas were prioritised by discussion and voting on three main objectives for each participating practice team (Practice A: better information towards patients by restructuring the waiting room, focusing on solution-oriented approach and lifestyle interventions; and Practice B: formulating a practice mission, giving special emphasis on lifestyle and a more efficient use of competences in the team). The teams agreed to re-evaluate their progress in three to five months. In the week after the facilitator session, a short report summarising the two sessions and the choices made was sent to the practices.

Two sessions (Box 1) were prepared and led by a professional practice facilitator of the regional facilitating organisation. The sessions were held in a common room within the practice during the working hours of the teams. The team members were paid by the practices. The practice facilitator was funded by the regional facilitating organisation. Filling in the survey was done via an online version of lime survey and was done in the individual's own time. The practice facilitator spent an average of 8.5 hours per participating practice team (0.5 hours explaining the project and making an appointment for the sessions, 2 hours preparing the sessions, 4 hours giving the sessions (2 hours per session), 2 hours collecting and analysing the surveys and creating a feedback report). The practice coordinator on behalf of the participating practice spent an average of 5.25 hours (0.5 hours explaining the project and making an appointment for the sessions, 4 hours after the sessions and, 0.75 hours completing the survey). The other team members spent an average of 4.75 hours (4 hours after the sessions and, 0.75 hours completing the survey). The average time investment in the two participating practices is a total of 60 hours per practice. In the evaluation interview after three months, the representative of practice A described a good feeling towards the process. At the start, everyone was dedicated, there were regular meetings evaluating the changes in daily practice, and changes were made in the scheduling of patient appointments to increase consultation time for certain patients. Some of the team members followed an educational course, "Solution-Oriented Approach". The rearrangement of the waiting room was not changed on financial grounds. Contacts had been made with physiotherapists and local social workers to arrange the coordination of lifestyle-changing interventions. In the long term, it appeared difficult to arrange all of the changes in daily practice and not every team member was equally committed to or willing to take responsibility. Another obstacle was the strictness of chronic care protocols, leaving little room for patient-centred initiatives. It is unclear what patients noticed in their daily care.

In practice B, some of the team members also underwent, or planned to, an educational course "Solution-Oriented Approach". In order to increase patient involvement, a survey on patient preferences and needs was sent out to all patients in the practice. Patients appeared to respond very enthusiastically, although preliminary results of the analyses were not available at the time of interview. Formulating a practice mission statement was still a work in progress.

## Discussion

### Principal findings

This paper described the development and a first pilot testing of the MM-CE in routine general practice, using a structured process and in accordance to the guidelines of the Medical Research Council on complex interventions [42]. Elements of care engagement gathered in a scoping literature search were tested and enriched by (focus group) interviews with stakeholders, as well as by scientific researchers and implementational consultants in primary care. In this way, we have designed a tool step by step that is targeted to care engagement. This tool was *evidence based* and *pragmatically* created in *partnership* with local stakeholders [42–44]. Due to its methodical and comprehensive development and based in the local setting, it was thought to embrace the broad field of patient engagement in a practical way. The prototype of the MM-CE distinguished seven domains, i.e., *Personalised Care*, *Shared Decision Making*, *Self-Management*, *Patient as Partner*, *Supportive Means*, *Patient Environment*, and *Teamwork among Healthcare Professionals*. Each of the domains was described in several activities (ranging from 3–6, with a total of 32). Two pilot practices followed an implementation protocol (kick-off session and facilitator session to support SMART improvement plans). The majority of these practice teams embraced the concept, although practice assistants had particular difficulty relating some of the items to their daily routines. After three months, some changes towards more care engagement were reported. The practices found it hard to keep the process on the right track over the long-term and thought that guidance for a longer period would be helpful.

### Practical and scientific implications

Higher care engagement or involving patients actively in healthcare contributes to quality of primary care [2, 3, 17, 18], but is not simple to implement in routine practice. There are recent studies on efforts to increase patient engagement [47–50]. However, to the best of our knowledge, tools are lacking to support activities to optimise the implementation of organisational aspects on care engagement in general practice. In cooperation with local stakeholders, we aimed to develop of a maturity matrix to be used in daily practice that could function as a starting point for change. Such a tool mapping the present state of care engagement in daily practice may provide transparency on the skills and means that are lacking and provide opportunities for the improvement of the organisation. From the evaluation and reflection on the implementation of the tool, we learned several things. The tool itself is not a finished product and will probably benefit from alteration being made. We will describe the lessons learned according to the seven aspects in the implementation framework of Flottorp [34].

**1. Tool-related factors.** The instrument was developed based on an extensive literature search in which we mapped all relevant aspects of patient engagement. Relevant stakeholders and experts grouped the items and adjusted the wording. However, some items need more alignment with the practice assistant's work. The organisations that have co-developed the instrument were recognised as reliable parties by the participating practices. However, it was still hard to recruit practices.

**2. Health professional factors.**   In both practices, some physicians have already expressed an interest in the concept of positive health and endorsed the benefits of patient engagement. This knowledge about positive health was not yet available to all employees. Post-intervention evaluations showed that practitioners endorsed the topic of the intervention and recognised the identified themes as being related to patient engagement.

**3. Patient factors.**   Our focus interviews showed that the patients did indeed feel the need to be more engaged. Therefore, the patients are behind this change. Even though there is some literature mentioning that patient engagement is not desirable in all circumstances. Patients may feel abandoned and left to their own fate, with not everyone being able or willing to be engaged and bear responsibility [22, 24].

**4. Professional interactions.**   The instrument was in line with the team's need for a movement towards greater patient engagement and with a general trend in health care in the Netherlands to be patient-centred. The tool aims to increase patient involvement across the whole practice team. Therefore, all team members were asked to complete the questionnaire before starting the discussion. However, it turned out that filling in the instrument took too long and, in that sense, was a considerable investment of time for the entire team. We suggest having the team meeting prepared by a facilitator and a limited number of team members (two to three) who complete the questionnaire. The results are then discussed by the entire team in a central meeting. In addition, it proved difficult for the practices to keep patient engagement on the agenda, especially the realisation of goals and the continuation of plans after the guided session. They expressed a strong need for guidance in this process. A more intensive deployment of practical facilitators may help in this regard. Furthermore, the practices were aware of the fact that changes in their practice do not stop at the boundaries of their practice and that they also need to interact with other parties in their vicinity, both in the field of healthcare and welfare. We suggest addressing this issue from the start.

**5. Incentives and resources.**   The local facility organisations have invested a lot of effort into this project by organising inspiration meetings, providing experienced practice facilitators (change experts) and offering an educational course. The questionnaires were easily available electronically to the general practices. Therefore, resources were easily available, and support was given by local organisations. The initial interest of 54 practices shows that the subject has the attention of many general practices. However, taking concrete follow-up steps by introducing the maturity matrix in general practice has lagged behind. The non-participating practices indicated that the timing of the study was not appropriate for them; these practices indicated that many extra tasks were already coming their way.

**6. Capacity for organisational change.**   In both practices, the primary care physicians were also the practice owners; as such, they had the mandate to participate in the intervention. Together with other stakeholders (such as the regional facilitating organisations) they felt a need to initiate the change towards more patient engagement. No real opponents emerged from the interviews and surveys afterwards, nor were there any rules or regulations opposing this. Nevertheless, it appeared difficult for practitioners to keep up with the chosen direction to change the practice. Practices specify that they need practical organisational support staff to reflect on the process from time to time and to keep things moving with the help of feedback.

**7. Social, political and legal factors.**   Although there is a lot of discussion about changing health care budgets in the Netherlands, existing contracts are mainly focused on providing day-to-day care, so there is little room to get budgets that focus on changing organisational processes. The government continues to pursue a policy in which budgets for primary care are even more reduced. In addition, the study took place at a time when practices were experiencing increased workloads due to the more difficult recruitment of staff and increased substitution from secondary care.

It might be concluded that the prototype is not yet fully developed. O'Cathain suggested that a prototype needs further refinement and optimalisation by a series of iterations in which an assessment takes the place of acceptably and feasibility [42]. The necessary refinements and optimalisation concern revising questions due to abstract language and ambiguity, but also team members should only be exposed to items relevant to them. This should make the instrument shorter and easier to use, thus increasing acceptance. Furthermore, a team meeting could be prepared by the facilitator with two or three individuals to complete the tool. Other maturity matrices exhibited good results with the method in which the whole practice team did not complete the tool [38, 40, 41]. We believe that this will not decrease team commitment and feedback will be discussed by the whole team in a central meeting. Finally, the practice plans established in the intervention should be supported for an adequate amount of time by the facilitators through practice visits, reminders and support materials. A well-supported post-trajectory is necessary for proper change support by practice facilitation and organisational leadership [9, 35]. The relevance of practical help, reminders and support tools was also found in research on the implementation of clinical guidelines in primary care [51]. These propositions will have to be discussed with stakeholders and tested in the next pilot. Changes in general practice need to be both relevant and easy to use to avoid detracting from the day-to-day business and service demands of patients [9, 35, 51]. We imagine that the MM-CE may be used as part of a toolbox facilitating the practice team in their change towards care-engagement, combined with additional complementary tools supporting implementation action and staying on track to set goals. The literature has shown that supporting a team with a toolbox helped to broaden the mindset regarding possible action that clinicians could take to change their daily practices [52].

## Strengths and limitations

One strength is the scoping review that defined the different aspects of care engagement. The focus groups with patients and healthcare professionals assisted in the interpretation and prioritisation of these different aspects, with experts confirming these choices and making several relevant remarks on the user guide. Another strength is that we were able to describe the whole process according to the Medical Research Council guideline for the development of complex interventions to improve health and healthcare [42]. It has been a thorough process taking time to combine theory, literature, and stakeholder input in order to define, refine and fine-tune a comprehensive tool to determine the level of care engagement of a general practice team. In the iterative process, the set of terms used is checked and double-checked in different settings before assembling it to the present tool. In order to increase acceptability, its implementation was modified in close consultation with local facilitating organisations to increase integration into regional policy and its feasibility. A limitation of this study is the number of participants, but the participating practices contributed actively.

## Conclusions

The present study described the systematic development of the maturity matrix of general practice organisations on care engagement. The existing literature was triangulated with the views of stakeholders. From the feasibility study, it was concluded that the tool was reasonably received in daily practice and has the potential to initiate changes towards increased care engagement. Adjustments of the instrument as well as implementation of the tool were suggested. The next step in the use of the maturity matrix on care engagement is to test an adapted version in a larger number of general practices.

## Supporting information

**S1 File. Scoping literature search.**
(PDF)

**S2 File. Elements in Maturity Matrix Care Engagement (MM-CE).**
(PDF)

## Acknowledgments

The authors wish to acknowledge the contributions made by Jasper Kortenhorst for assistance with data preparation and analysis. We appreciate Glyn Elwyn and Marjan Meinders for their critical reflections of the concept tool. We further thank Marion Reinaerdz from *Zorgbelang Gelderland*, Simone Kwant and Clary te Velde from *Onze Huisartsen* and Chantal Walg, Rudie van den Berg and Monique Schmidt from *Proscoop* for supporting the focus group interviews, discussions in the shaping of the MM-CE and eventual facilitation of testing of the feasibility of the MM-CE. We are also thankful to general practice team members of *Glux* and *Dokter-shuis* for the willingness to test the tool in their daily work.

## Author Contributions

**Conceptualization:** René Wolters, Ibtissam Mokadem, Jozé Braspenning.

**Data curation:** René Wolters, Ibtissam Mokadem, Jozé Braspenning.

**Formal analysis:** René Wolters, Ibtissam Mokadem, Michel Wensing, Jozé Braspenning.

**Funding acquisition:** Michel Wensing, Jozé Braspenning.

**Investigation:** René Wolters, Ibtissam Mokadem.

**Methodology:** René Wolters, Ibtissam Mokadem.

**Project administration:** Ibtissam Mokadem.

**Supervision:** Jozé Braspenning.

**Writing – original draft:** René Wolters.

**Writing – review & editing:** René Wolters, Ibtissam Mokadem, Michel Wensing, Jozé Braspenning.

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
