## [Decision Letter · Decision Letter 0]

22 Jun 2022

PONE-D-21-38853

Supporting care engagement in primary care; the development of a maturity matrix

PLOS ONE

Dear Dr. Wolters,

Thank you for submitting your manuscript to PLOS ONE. After careful consideration, we feel that it has merit but does not fully meet PLOS ONE’s publication criteria as it currently stands. Therefore, we invite you to submit a revised version of the manuscript that addresses the points raised during the review process.

Please note that we have only been able to secure a single reviewer to assess your manuscript. We are issuing a decision on your manuscript at this point to prevent further delays in the evaluation of your manuscript. Please be aware that the editor who handles your revised manuscript might find it necessary to invite additional reviewers to assess this work once the revised manuscript is submitted. However, we will aim to proceed on the basis of this single review if possible. 

Although there was only one reviewer, that person has provided many useful and detailed suggestions - please see the attached document.

We look forward to receiving your revised manuscript.

Kind regards,

Steve Zimmerman, PhD

Associate Editor, PLOS ONE

**Journal requirements:**

“This study was funded by an unconditional grant from AQUA-Institut, Göttingen, Germany.”

“This study was funded by an unconditional grant from AQUA-Institut, Göttingen, Germany. (https://www.aqua-institut.de/) The funder played no role in the study design, data collection and analysis, decision to publish, or preparation of the manuscript”

Reviewers' comments:

Reviewer's Responses to Questions

**Comments to the Author**

1. Is the manuscript technically sound, and do the data support the conclusions?

Reviewer #1: Partly

2. Has the statistical analysis been performed appropriately and rigorously? 

Reviewer #1: N/A

3. Have the authors made all data underlying the findings in their manuscript fully available?

Reviewer #1: Yes

4. Is the manuscript presented in an intelligible fashion and written in standard English?

Reviewer #1: Yes

5. Review Comments to the Author

Reviewer #1: Please see uploaded review. I think this the start of a good paper - more work is needed to enable it to be in publishable form. Bringing in implementation science principles and fleshing out the discussion will be necessary.

6. PLOS authors have the option to publish the peer review history of their article (what does this mean?). If published, this will include your full peer review and any attached files.

Reviewer #1: **Yes: **Rajna Ogrin

---

## [Author Response · Author response to Decision Letter 0]

18 Sep 2022

Dear Dr Zimmerman, Dear Editors,

We would like to thank you for giving us the opportunity to submit a revised version of our manuscript titled “Supporting care engagement in primary care; the development of a maturity matrix” (PONE-D-21-38853). We also would like to thank the reviewer for her generous and helpful comments on the manuscript.

In the additional ‘point to point rebuttal’ we discussed the points raised. We have included multiple links throughout the paper to the appropriate code repositories. 

We have checked the requirements of the journal and adapted the paper to it. Furthermore, we discussed making the data available for a larger audience with a member of the ethical committee of the Radboudumc. Based on this discussion we provide a new statement on Availability of data and materials: “Participants have given their informed consent for the use of anonymized fragments of qualitative data. Participants did not consent to provision of the full raw dataset to persons other than the research team. Since the raw interviews and transcripts contain sensitive information, even anonymized raw data can compromise their confidentiality. Therefore, current Dutch privacy law and institutional regulations prevent the a priori sharing of the full raw dataset. Considering the importance of data-sharing and providing insight into the research, data access will be considered upon request, evaluating each inquiry individually. Requests for data access may be sent to the department of IQ healthcare of the Radboudumc at iqhealthcare@radboudumc.nl.” 

And at your request we include now the amended funding statements in the cover letter: 

“This study was funded by an unconditional grant from AQUA-Institut, Göttingen, Germany. (https://www.aqua-institut.de/) The funder played no role in the study design, data collection and analysis, decision to publish, or preparation of the manuscript”. 

We agree with the proposed update of the funding information in the Funding Statement section of the online submission form.

We believe that the manuscript is now suitable for publication in PLOS ONE.

Yours sincerely,

René Wolters, MD PhD,

General Practitioner, Researcher

On behalf of the co-authors. 

Point to point rebuttal

General:

• Please go over the paper and correct any sentences that need grammar or sentence structure improvement. Also to include end brackets and other little details that have been missed.

We have checked the document for errors and have made corrections throughout.

Abstract

Background:

• First sentence is too vague. “Care engagement or active patients’ involvement in healthcare may contribute to the quality of primary care” – I think you need to be clear that increased engagement may improve quality of care.

• Second sentence doesn’t seem to link to first sentence – suggest need to amend to something like: maturity matrix considers organisational preconditions, and we developed and evaluated it… etc…

Thank for your comment, we have rephrased the first two sentences (lines 22-29 in 'Revised Manuscript with Track Changes’ ).

Introduction 

• Please change your language around older people – ‘elderly’ is not appropriate. Suggest you look at eg. From journals focused on research in older people for guidance: When It Comes to Older Adults, Language Matters: Journal of the American Geriatrics Society Adopts Modified American Medical Association Style - Lundebjerg - 2017 - Journal of the American Geriatrics Society - Wiley Online Library 

Yes, we do agree with your suggestion and have amended the entire text accordingly.

• Page 4, line 71 – person-centred care does not enable ‘compliance’ – in fact, that is the opposite of person-centred care. Suggest reframe. 

We agree. The word “compliance” has been removed (line 79).

 

• I wonder if the authors need to consider some papers on interprofessional collaborative care that focuses on team collaboration, where the person seeking healthcare is at the centre eg. Interprofessional.Global and Framework for action on interprofessional education & collaborative practice (who.int) and IPEC Core Competencies (ipecollaborative.org) and ASPPH | Interprofessional Collaborative Practice Model 

Interprofessional collaborative care is indeed an important aspect of proper patient involvement. This was one of the main reasons why we worked on the development of our tool. We added a line on this subject with some relevant references (lines 64-65).

• I also wonder whether you need to bring in implementation science principles here – you state in page 5, line 77 “To translate this scientific information…’ – this is a field of science with underpinning core components. You are endeavouring to implement this into practice, so suggest you need to introduce what that means a bit more eg. An implementation framework that seems to fit what you are doing = CIFR The Consolidated Framework for Implementation Research – Technical Assistance for users of the CFIR framework (cfirguide.org)

Our initial intention was to describe the development of the maturity matrix as a tool according to the principles of the development of a complex intervention (O’Cathain, 2019). We agree with our reviewer that in the introduction the implementation aspects of are underexposed. Extra lines were added (lines 85 to 90). Furthermore, the Medical Research Counsil guideline for the development of complex intervention is addressing implementation aspects. We added this point (line 120). Moreover, we now discussed our lesson learned in the discussion according to the implementation framework of Flottorp et al (2013), lines 415-509.

• Page 5, line 91 – the plural of matrixes is matrices

• Please review the grammar and language in the introduction – some tweaking needed eg. Sentence starting page 5, line 93 – think ‘a’ is messing after the first word ‘Such’, with that sentence being a bit clunky

The text is modified accordingly (lines 105 and 107)

Methods

Patient population - Broad inclusion criteria. Any exclusion criteria?

There were initially no exclusion criteria for the focus group interviews. The recruitment of patients and care professionals was by purposive sampling in order to get a certain diversity in the panels. The text in the Methods has been adapted accordingly in order to provide more information on this (lines 174-185)

 

Outcome measures - What about the resources needed to do this work as part of the feasibility and acceptability of undertaking this work? Ie. How long do people need to spend to do preparation, the actual filling out of the matrix, meeting with others etc..? Do they get back filled to do this work? Does it cost anything to do? Can they do it on their own or do they need get a facilitator in? Who pays for that? Etc..

We really appreciate your suggestions for extra information. Outcome measures were gathered by post intervention enquiries (Line 221). Outcome as such has been presented in de Results Section (lines 337-349)

Results 

Table 1

• Patient participants – think we need to know their age, their education level, their socio-economic information, cultural background, other? – need a better picture of who they are to help us know from which perspective they are coming from;

• Patient organisations – are they not for profit? Government funded? How long have they been in operation? What size are they? Etc..

• Care professionals – age, education level, how long they have been working in total, and how long in the current role 

• For all these participants – are they representative of their respective groups?

Unfortunately we don’t have information on educational level, socio-economic and cultural background of all the interviewees. The participants were purposive sampled in order to get a certain case mix. In the selected patients their age, gender and medical condition were leading. The care professionals sampled to get a certain mix in age, number of years’ experience, gender and the urbanisation level they worked. The information we did have in our possession we now have added to table 1 and lines 174-185.

Table 2

• Need more info on the GPs, practice nurses etc… age, education undertaken, how long they have been working and how long they were in the current role.

We provided additional information in the main text (Lines 286, 287 and 290-295).

• I wonder – what about receptionists being involved in something like this? They talk a lot with patients, and often know a lot about them…

In the Netherlands we don’t have a separate receptionist. The “practice assistant” performs receptionist tasks as well as simple delegated and protocolled medical procedures (such as measuring blood pressure, sampling blood, injecting medication, rinsing cerumen and removing stitches). We have therefore provided explanatory text in the footnote of Table 2 and in the main text in the lines 293-295

 

• Is there any difference in these 2 practices that participated compared to the other 50 or so who had been sent invitations?

The two participating practices are in their setting (practice size, number of care professionals, level of urbanisation and social economic setting of their neighbourhood) considered as examples for the predominant practice in the region. (line 286-287).

Discussion

• Suggest you start off with the core of what you found rather than the issue you are addressing.ie. Start with the sentence page 16 line 318 “This paper described the development as well as a first pilot…”

Thanks very much for this suggestion. Text becomes clearer. We removed these lines, and started the discussion as suggested (lines 369-373).

• Suggest your current first sentence needs reframing – “care engagement is thought to contribute to the quality of primary care”. You mean high care engagement = higher quality of care, so I suggest that is articulated.

This text has been removed from the first paragraph and is now present in line 406 to articulate this point

• This discussion simply states what you did, and doesn’t relate it back into the existing literature – has anyone else done this kind of thing before? What did they find? How did your findings differ to what was done before? How does your work add to the knowledge base in this area? 

• What about implementation aspects? This took considerable effort on the part of the research team to set up with the practices to implement – what would be needed for every general practice to do this? How would that even look? Is that even feasible? If this is an implementation study, need to consider what is needed to have this activity as part of ‘Business as Usual’ – as per my comment in the introduction, I strongly recommend you look up implementation science frameworks re: what is needed. A good starting point is reporting of implementation studies: Pinnock H, Barwick M, Carpenter CR, Eldridge S, Grandes G, Griffiths CJ, Rycroft-Malone J, Meissner P, Murray E, Patel A, et al. Standards for Reporting Implementation Studies (StaRI): explanation and elaboration document. BMJ Open. 2017;7:e013318. Things to consider – could practices do this on their own, or do they need a facilitator? How long would it take? Who would pay for backfill? What supports would be needed? What is needed to actually work through the practice changes identified from this process? 

• One thing that also struck me – the groups came up with what they need to change, but struggled to make these changes. That must surely be somewhat anxious making for all participants – no one wants to fail. What is the impact of this inability to implement the changes they identified they needed? This can cause moral distress when care providers can’t deliver the best care they know they should. 

Strengths and limitations

• You mention the low uptake to be a test practice – I suggest the authors response that it is due to limited interest is insufficient and would recommend reading the implementation science literature around the multitude of structural reasons behind limited uptake. This requires more comprehensive information – suggest asking a few of those who were approached but declined would also be important. 

Practical and scientific implications

• Ah – now I see you bringing in other literature, thank you.

• I wonder – of the tool, what are the essential components to promote your goal – it looks like there are many components which takes a lot of time and input – is it all required? I think you do need to engage all practice team members – but do they need to do the same as the others? Should they look at things a bit differently, given their role and the skills they have are different? OR do you need to modify what you are considering so that you are only including those aspects relevant to all team members? What about receptionists? As I mentioned above, they know patients well – how can they be included? They are the first port of call to the practice, and can be barriers or enablers to patients feeling welcome and psychologically safe…

• You talk about raising enthusiasm alone is not enough – this leads into the implementation science approaches that need to be considered. Suggest more is needed here to support moving forward. The paragraph starting page 18 line 367 requires reworking – you wend in practice changes required by healthcare practices, then bring in the big issues around those who seek care and those factors that may limit their capability to engage ie. Health literacy, cognitive impairment etc… This needs more explanation. Right now, you have raised these topics, and they are very complex and you have just skimmed the surface.

• The last sentence starting line 381 is very long and convoluted - suggest reframing. Need to outline that the findings need to be used to adapt the current tool and test to ensure it meets the end user needs. Right now it is long, involved, not all team members can do all of it and it doesn’t lead to practice change. Sorry to be so blunt! But this is not a bad thing – it just means that this is where you have to focus your next steps – how can you make it include all key stakeholders? How can it be less time consuming – what can you leave out? What are the bits you simply must keep in? What are the essentials that could lead to real practice change? What supports are needed so practices can undertake practice change?

We really appreciate your comments on our discussion. We therefore reformulated our entire discussion focused on the lessons learned accordingly to the implementation framework of Flottorp (2013) (lines 406 to 531).

Conclusions

I think this tool requires significant adaptations to enable it to be used in practice, and that working out what is essential and how practice change can be supported are key. 

Conclusions have been rephrased in accordance with this suggestion and the changes made to the text (lines 532-539).

---

## [Decision Letter · Decision Letter 1]

6 Nov 2022

PONE-D-21-38853R1Supporting care engagement in primary care; the development of a maturity matrixPLOS ONE

Dear Dr. Wolters,

Thank you for submitting your manuscript to PLOS ONE. After careful consideration, we feel that it has merit but does not fully meet PLOS ONE’s publication criteria as it currently stands. Therefore, we invite you to submit a revised version of the manuscript that addresses the points raised during the review process.

We look forward to receiving your revised manuscript.

Kind regards,

Filipe Prazeres, MD, MSc, Ph.D.

Academic Editor

PLOS ONE

Journal Requirements:

Reviewers' comments:

Reviewer's Responses to Questions

**Comments to the Author**

1. If the authors have adequately addressed your comments raised in a previous round of review and you feel that this manuscript is now acceptable for publication, you may indicate that here to bypass the “Comments to the Author” section, enter your conflict of interest statement in the “Confidential to Editor” section, and submit your "Accept" recommendation.

Reviewer #1: (No Response)

Reviewer #2: (No Response)

2. Is the manuscript technically sound, and do the data support the conclusions?

Reviewer #1: Yes

Reviewer #2: Yes

3. Has the statistical analysis been performed appropriately and rigorously? 

Reviewer #1: N/A

Reviewer #2: Yes

4. Have the authors made all data underlying the findings in their manuscript fully available?

Reviewer #1: No

Reviewer #2: Yes

5. Is the manuscript presented in an intelligible fashion and written in standard English?

Reviewer #1: No

Reviewer #2: No

6. Review Comments to the Author

Reviewer #1: Thank you for considering the feedback and making the changes you did. It is more comprehensive and reads much better. It is a good study and would be of interest to others researching and working to improve person-centred care delivery in general practice.

General: There is still some grammar and sentence structure improvement required.

Re: discussion;

You described the lessons learned using the seven aspects from Flottorp et al’s Implementation Framework – I am wondering whether this should be part of the results. I see this as an integral part of implementation evaluation – I don’t know that it fits in the discussion. If you agree, I think you would need to add that you use this framework to evaluate the implementation into your methods.

Reviewer #2: General:

• There is an inconsistent spelling of certain words throughout the manuscript (e.g., organisational / organizational, and Personalised / Personalized have been used interchangeably)

• Please consider copy-editing this manuscript. There are several grammar, spacing, spelling, and punctuation errors (the most significant being the overuse of unnecessary commas), alongside unclear antecedents and dangling modifiers. Appropriate quality copyediting should remove these errors. I have listed some errors I found, but there are several more that need attention.

Abstract

• Line 23 has a grammatical error (e.g. preconditions instead of preconditions)

Background:

Introduction

• The sentence spanning from lines 72-75 is too long (49 words). I know it is a quote, but Please consider splitting it into two or paraphrasing it into a shorter sentence.

• Line 76: It is not good practice to use unclear antecedents like ‘this’ to begin a sentence. It may be unclear what ‘this’ refers to. Consider re-writing the sentence to remove the unclear reference.

• Line 78: contains a dangling modifier (i.e. “To implement these insights on care engagement in a sustainable manner” does not appear to be modifying the subject). Sentence rearrangement should clarify this.

Methods

• The sentence spanning lines 189-191 needs rewriting for clarity. Some suggestions:

• Appropriate writing conventions require authors to write out any numbers under ten in full (e.g., ‘two’ instead of ‘2’)

• “of whom 2 implementation consultants” is missing a word.

• The sentence is too long and unclear.

• The sentence spanning lines 202-203 needs rewriting for clarity, and an end bracket.

• The sentence spanning lines 205-206 has a spelling error (where VS were). I suggest fixing.

Results

• Line 221: A total of 38 papers were considered relevant for the identification of the axis. This should be reworded, as there is overcomplexity and redundancy in the sentence formulation. (i.e., A total of 38 papers were considered relevant for axis identification). Sentence overcomplexity is a very common issue throughout this manuscript.

Discussion

• The sentence spanning lines 350-351 “This way we designed in a stepped way a target population centred tool.” Needs to be addressed for punctuation, grammar and clarity.

• There are several instances of incorrect use of conjunctions, The sentence on line 358 is an example, (kick-off session, facilitator session to support SMART improvement plans). Please add appropriate conjunctions (e.g., ‘and’ where appropriate).

• There are several instances of incorrect word spacing throughout the manuscript. The sentence on line 359 is an example.

• There are several instances of incorrect, or lacking, punctuation use. The sentence on lines 365-368 is an example (Although there are recent studies on efforts to increase patient engagement, to our knowledge there is no existing tool…) I suggest to add in comma handles where appropriate to this sentence. Please also review the manuscript to remove excessive or add lacking punctuation. The sentence on lines 398-400 is also an example of lacking punctuation.

7. PLOS authors have the option to publish the peer review history of their article (what does this mean?). If published, this will include your full peer review and any attached files.

Reviewer #1: **Yes: **Rajna Ogrin

Reviewer #2: **Yes: **Harry James Gaffney

---

## [Author Response · Author response to Decision Letter 1]

24 Nov 2022

Point to point rebuttal regarding reviewers comments to the author

Reviewer #1: Thank you for considering the feedback and making the changes you did. It is more comprehensive and reads much better. It is a good study and would be of interest to others researching and working to improve person-centred care delivery in general practice.

General: There is still some grammar and sentence structure improvement required.

• Following your suggestion, we have had the manuscript checked by a text editor. 

Re: discussion; 

You described the lessons learned using the seven aspects from Flottorp et al’s Implementation Framework – I am wondering whether this should be part of the results. I see this as an integral part of implementation evaluation – I don’t know that it fits in the discussion. If you agree, I think you would need to add that you use this framework to evaluate the implementation into your methods.

• We understand the consideration, but the focus of our paper is on supporting care engagement in primary care for which we have developed a tool. We endorse the importance of taking implementation into account from the start. That is why we have systematically approached the development of the tool (see also the Medical Research Council guideline for the development of complex interventions) and have involved the local stakeholders from the start. Flottrop’s Implementation Framework was introduced after the evaluation of the study. We therefore don’t think it is methodologically sound to suggest this was already part of the original plan. However, we do value discussing our results in the context of this framework, as the framework makes the lessons learned more concrete.

Reviewer #2: General:

• There is an inconsistent spelling of certain words throughout the manuscript (e.g., organisational / organizational, and Personalised / Personalized have been used interchangeably)

• Please consider copy-editing this manuscript. There are several grammar, spacing, spelling, and punctuation errors (the most significant being the overuse of unnecessary commas), alongside unclear antecedents and dangling modifiers. Appropriate quality copyediting should remove these errors. I have listed some errors I found, but there are several more that need attention.

Grammatical suggestions form reviewer 2

Abstract

• Line 23 has a grammatical error (e.g. preconditions instead of preconditions)

Introduction

• The sentence spanning from lines 72-75 is too long (49 words). I know it is a quote, but Please consider splitting it into two or paraphrasing it into a shorter sentence.

• Line 76: It is not good practice to use unclear antecedents like ‘this’ to begin a sentence. It may be unclear what ‘this’ refers to. Consider re-writing the sentence to remove the unclear reference.

• Line 78: contains a dangling modifier (i.e. “To implement these insights on care engagement in a sustainable manner” does not appear to be modifying the subject). Sentence rearrangement should clarify this.

Methods

• The sentence spanning lines 189-191 needs rewriting for clarity. Some suggestions:

• Appropriate writing conventions require authors to write out any numbers under ten in full (e.g., ‘two’ instead of ‘2’)

• “of whom 2 implementation consultants” is missing a word.

• The sentence is too long and unclear.

• The sentence spanning lines 202-203 needs rewriting for clarity, and an end bracket.

• The sentence spanning lines 205-206 has a spelling error (where VS were). I suggest fixing.

Results

• Line 221: A total of 38 papers were considered relevant for the identification of the axis. This should be reworded, as there is overcomplexity and redundancy in the sentence formulation. (i.e., A total of 38 papers were considered relevant for axis identification). Sentence overcomplexity is a very common issue throughout this manuscript.

Discussion

• The sentence spanning lines 350-351 “This way we designed in a stepped way a target population centred tool.” Needs to be addressed for punctuation, grammar and clarity.

• There are several instances of incorrect use of conjunctions, The sentence on line 358 is an example, (kick-off session, facilitator session to support SMART improvement plans). Please add appropriate conjunctions (e.g., ‘and’ where appropriate).

• There are several instances of incorrect word spacing throughout the manuscript. The sentence on line 359 is an example.

• There are several instances of incorrect, or lacking, punctuation use. The sentence on lines 365-368 is an example (Although there are recent studies on efforts to increase patient engagement, to our knowledge there is no existing tool…) I suggest to add in comma handles where appropriate to this sentence. Please also review the manuscript to remove excessive or add lacking punctuation. The sentence on lines 398-400 is also an example of lacking punctuation.

• Thank you for carefully reading our contribution. We have adapted the passages you have designated. In addition, we also had the text reviewed by a professional editor, as you suggested.

---

## [Editor Report · Decision Letter 2]

12 Dec 2022

Supporting care engagement in primary care; the development of a maturity matrix

PONE-D-21-38853R2

Dear Dr. Wolters,

We’re pleased to inform you that your manuscript has been judged scientifically suitable for publication and will be formally accepted for publication once it meets all outstanding technical requirements.

Kind regards,

Filipe Prazeres, MD, MSc, Ph.D.

Academic Editor

PLOS ONE
---

## [Editor Report · Acceptance letter]

23 Dec 2022

PONE-D-21-38853R2 

Supporting care engagement in primary care; the development of a maturity matrix 

Dear Dr. Wolters:

I'm pleased to inform you that your manuscript has been deemed suitable for publication in PLOS ONE. Congratulations! Your manuscript is now with our production department. 

Kind regards, 

on behalf of

Prof. Filipe Prazeres 

Academic Editor

PLOS ONE